# Potential Use of *Ascophyllum nodosum* as a Biostimulant for Improving the Growth Performance of *Vigna aconitifolia* (Jacq.) Marechal

**DOI:** 10.3390/plants10112361

**Published:** 2021-11-02

**Authors:** Nidhi Verma, Krishnan D. Sehrawat, Poonam Mundlia, Anita R. Sehrawat, Ravish Choudhary, Vishnu D. Rajput, Tatiana Minkina, Eric D. van Hullebusch, Manzer H. Siddiqui, Saud Alamri

**Affiliations:** 1Department of Botany, Maharshi Dayanand University, Rohtak 124001, India; nidhi.verma40@gmail.com; 2Department of Genetics and Plant Breeding, CCS Haryana Agricultural University, Hisar 125004, India; krishanssehrawat@gmail.com; 3Department of Biochemistry, Punjab University, Chandigarh 160014, India; poonammundlia@gmail.com; 4Division of Seed Science and Technology, ICAR-Indian Agricultural Research Institute, New Delhi 110012, India; ravianu1110@gmail.com; 5Academy of Biology and Biotechnology, Southern Federal University, 344090 Rostov-on-Don, Russia; rvishnu@sfedu.ru (V.D.R.); tminkina@mail.ru (T.M.); 6Institut de Physique du Globe de Paris, Université de Paris, CNRS, F-75005 Paris, France; vanhullebusch@ipgp.fr; 7Department of Botany and Microbiology, College of Science, King Saud University, Riyadh 11451, Saudi Arabia; mhsiddiqui@ksu.edu.sa (M.H.S.); saualamri@ksu.edu.sa (S.A.)

**Keywords:** *Vigna aconitifolia*, pot foliar application, pot root application, *Ascophyllum nodosum* extract

## Abstract

The fertilizers that are derived from seaweed are known as Seaweed Liquid Fertilizers (SLF). SLF is a modern, cheap, non-toxic, and natural bioactive fertilizer. Among different studied seaweeds, *Ascophyllum nodosum* is significant as having bioactive ingredients that potentially regulate the molecular, physiological, and biochemical processes of crop plants. In the present study, the effects of the application of different concentrations (0.00%, 0.01%, 0.02%, 0.05%, 0.10%, 0.50%, and 1.00%) of *A. nodosum* Extract (ANE) to the *Vigna aconitifolia* through roots (Pot Root Application, PRA) and on the leaves (Pot Foliar Application, PFA) were monitored via the plant growth. The lower concentrations of ANE in both the PRA and PFA experiments showed positive growth on *V. aconitifolia.* The 0.10% ANE stimulated the maximum shoot growth when applied through the roots, while 0.05% ANE in both PFA and PRA experiments led to an increase in the number of pods, nodules, organic content, and moisture percentage. The 0.10% ANE also increased the leaf numbers, leaf area, and photosynthetic pigments. Hence, the application of 0.05% and 0.10% of *A. nodosum* extract in two ways (i.e., Pot Foliar Application, PFA, and Pot Root Application, PRA) ameliorated the growth capabilities of *V. aconitifolia*.

## 1. Introduction

Modern agriculture practices involve a lot of pesticides and fertilizers in order to fulfill the food needs of exponentially expanding populations. The uncontrolled accumulation of pesticides in the food chain leads to serious complications [1]. Sustainable, natural biocides and biostimulants that could increase the productivity without negatively affecting the environment are of paramount importance. The marine ecosystem is a vast natural resource [2]. Among the three groups (Phaeophyceae, brown; Rhodophyceae, red; and Chlorophyceae, green), brown seaweeds, are the most commonly used biofertilizers in agriculture [3]. Brown algae is a modern, cheap, non-toxic, and natural bioactive fertilizer [4]. It contains essential components which promote growth and yield [5]. It enhances resistance against biotic and abiotic stresses [6,7,8,9]. Brown algae accelerates seed germination, and improves the biomass as well as the moisture content of *Vigna radiata* [10]. Seaweed fertilizer is an effective alternative to chemical fertilizer, being easily absorbable by plants, and has no harmful effects on the ecosystem [11]. Alginates, fucoidan, laminarin, mannitol polyphenol, polysaccharides, etc., in brown seaweeds help in soil conditioning, moisture retention, aeration, and nutrient adsorption, ultimately resulting in enhanced soil fertility [12]. *A. nodosum* has taken a priority place in agriculture due to its extraordinary capacity to improve crop growth [13]. *A. nodosum* extracts (ANE) are commercially available in the market, with various trade names such as Kelprosoil, Biovita, Acadian, Alg-A-Mic, Nitrozime, Bio-Genesis, High Tide, Guarantee, Espoma, Kelp Meal, Agri-Gro Ultra, Maxicrop, Soluble Seaweed Extract, Stimplex, Synergy, etc. [14]. Commercial extracts of *A. nodosum* increase the antioxidant activity, refs [15,16] total phenolic and flavonoid contents, root and shoot growth, root colonization, and root nodulation [17,18] in crop plants. Protein-rich legumes are a major source of food in a vegetarian diet. They contain a good number of dietary fibers and micronutrients [19]. The active ingredients in beans are associated with health benefits and aid in the prevention of diseases [20]. Today, organic farming is proving to be successful as a therapy to the evil of modern chemical agriculture, and seaweed extracts can be an interesting alternative. The present research was an attempt to study the effect of different concentrations of *A. nodosum* extract on some growth indices and yield attributes of *V. aconitifolia*.

## 2. Results

### 2.1. Ascophyllum Nodosum Extract (ANE) Treatment and Seed Germination

The *Ascophyllum nodosum* extract (ANE) at 0.00% (control), 0.01%, 0.02%, 0.05%, 0.10%, 0.50%, and 1.00% effected seed germination of *V. aconitifolia*. It was decreased with the increasing concentration of ANE, and reduced even lower than that of control at high concentration (0.50%) of ANE. Almost no germination was observed when the seeds were treated with 1.0% ANE (Figure 1). The percentage of seed germination increased up to 0.05% ANE treatment and decreased thereafter. The maximum germination percentage (96.67 ± 5.77) was obtained at 0.01% ANE treatment on the third day, and that was significantly different from that of the control (ANOVA; *p* < 0.01, Table 1). The germination percentage on the fifth day ranged from 66.67 ± 15.28 to 3.33 ± 5.77 at 0.00 to 0.10% of the ANE treatment groups. The seed germination was increased until the seventh day and the maximum germination of seeds took place on the third day (ANOVA; F = 23.32; *p* < 0.0001; Table 1). Seaweeds have been reported to positively modulate the germination in *Lactuca sativa*, *Brassica oleracea*, and *V. radiata* [21,22].

### 2.2. Efficacy of ANE Treatments on Shoot and Root Lengths of V. aconitifolia

The shoot length was recorded every 15 days for a total of 75 days in both pot foliar application (PFA) and pot root application (PRA) of 0.00%, 0.01%, 0.02%, 0.05%, 0.10%, and 0.50% ANE (Figure 2, Appendix A). The shoot length increased exponentially up to the 60th day. PFA treatment of ANE showed longer shoot length than when applied through roots. The maximum shoot growth was recorded at 0.10% concentration of ANE in both PFA and PRA treatment.

The maximum shoot length (11.04 ± 1.95, 13.63 ± 1.68, 17.39 ± 2.21, 21.66 ± 2.34, and 22.40 ± 1.73 cm on the 15th, 30th, 45th, 60th, and 75th days, respectively) was observed when 0.10% ANE was applied through the roots. While after 0.10% ANE treatment, when applied via foliar application, the shoot length was recorded as 12.67 ± 1.41, 14.58 ± 2.09, 19.59 ± 3.37, 25.04 ± 3.78, and 26.20 ± 03.69 on the 15th, 30th, 45th, 60th, and 75th days, respectively. Significant difference in shoot growth was observed on the 15th day in PRA treatment (ANOVA; F = 18.80; *p* < 0.0001) and on the 60th day (ANOVA; F = 6.60; *p* < 0.01). The differences were non-significant in the PFA experiment (Table 2). In both the PFA and PRA experiments, 0.05% ANE treatment showed the greatest root length (Figure 3). The root length was 13.30 ± 0.52 and 17.40 ± 1.40 cm in PFA and PRA, respectively, at 0.05% ANE treatment. The growth was significantly higher than that of the control groups (HSD Tukey; *p* < 0.01 each). The root length ranged in PFA from 05.70 ± 2.17 to 13.30 ± 0.52 cm (ANOVA; F = 10.6; *p* < 0.01), and in PRA from 08.80 ± 0.45 to 17.40 ± 1.40 cm (ANOVA; F = 11.33; *p* < 0.01), respectively (Table 3). In both experiments (PFA and PRA), the minimum root length was recorded in control plants. Greater root length was recorded when the treatment was given via the roots compared to that of the foliar application. The ANE via root application was more evident on 15th day. Root application of ANE was found to be more effective than foliar application.

### 2.3. Effect of ANE Treatments on Leaf Number and Leaf Area of V. aconitifolia at 30th Day

The number of leaves was counted on the 30th day in both PFA and PRA experiments (Figure 3). The number of leaves was approximately equal in both the experiments (T Test; NS). Both foliar and root application of 0.50% ANE treatment decreased the number of leaves. The maximum number of leaves was found at 0.10% ANE treatment in both experiments.

The maximum number of leaves, 12.50 ± 1.13 in PFA (ANOVA; F = 8.43; *p* < 0.0001) and 13.00 ± 1.65 in PRA (ANOVA; F = 12.31; *p* < 0.0001, Figure 3; Table 4), was observed. The leaf area was different when ANE was given as PFA and PRA (ANOVA; F = 1.68; *p* = 0.21) and PFA (ANOVA; F = 1.92; *p* = 0.16; Table 5). The leaf area was greatest at 0.10 % ANE treatment. The root application of ANE showed a greater leaf area (12.12 ± 3.99 cm^2^) as compared to when ANE was applied via foliar application (7.20 ± 1.90 cm^2^) (Figure 3). The number of leaves and leaf area increased with increasing concentrations of ANE and an increase was noticed up to 0.10% ANE.

### 2.4. Effect of ANE Treatments on Nodulation

The nodules in ANE-treated *V. acontifolia* were large, round, pinkish, and clustered, compared to the control which were small, light brownish, and in single nodules. This was observed in both (PFA and PRA) experiments. The 0.05% ANE treatment resulted in the most effective concentration for nodulation (Figure 4).

### 2.5. Effect in Pot Foliar Application (PFA) and Pot Root Application (PRA) of ANE on Biomass Accumulation

The fresh weight and dry weight of the shoots and roots were recorded to calculate the organic content. The organic content of the roots in PFA were 0.05 ± 0.01 to 0.28 ± 0.01 (ANOVA; F = 10.7; *p* < 0.01) and for PRA were 0.10 ± 0.12 to 0.37 ± 0.04 (ANOVA; F = 21.83; *p* < 0.01). The moisture content of the shoots was 0.61 ± 0.04 to 4.41 ± 0.15 (ANOVA; F = 51.30; *p* < 0.01) for PFA and 0.50 ± 0.07 to 6.71 ± 0.46 (ANOVA; F = 51.60; *p* < 0.01) for PRA application of ANE (Figure 5, Table 6 and Table 7). The moisture content of PFA and PRA in the roots was 0.19 ± 0.03 to 0.46 ± 0.05 (ANOVA; F = 4.0941; HSD Turkey = NS) and 0.25 ± 0.06 to 0.36 ± 0.06 (ANOVA; F = 3.01; HSD Tukey = non-significant), respectively. The moisture percentages of the shoots in PFA were 69.20 ± 1.20 to 85.22 ± 0.86 (ANOVA; F = 8.66; *p* < 0.01) and 52.73 ± 10.60 in PRA to 84.47 ± 0.97 (ANOVA; F = 6.64; *p* < 0.01) (Figure 5). The moisture percentage of the roots in PFA and PRA were 38.30 ± 2.44 to 84.54 ± 1.83 (ANOVA; F = 3.54; *p* < 0.05) and 45.73 ± 5.40 to 69.78 ± 5.85 (ANOVA; F = 7.11; *p* < 0.05), respectively (Table 6 and Table 7). In both PFA and PRA experiments, the organic content, moisture content, and the moisture percentage were greatest at 0.10% ANE treatment.

### 2.6. Effect of ANE on Number of Pods and Seed Yield

The number of pods in PFA ranged from 1.30 ± 0.57 to 4.30 ± 0.57 (ANOVA; F = 12.2; *p* < 0.01), whereas in PRA 2.00 ± 1.00 to 5.00 ± 1.70 (ANOVA; F = 4.4; *p* < 0.05; Table 8, Figure 6) were observed. The greatest number were recorded at 0.10% ANE treatment, through the roots.

### 2.7. Efficacy of ANE Treatments on Photosynthetic Pigment Accumulation

Photosynthetic pigment accumulation increased with increasing ANE concentration and reached a maximum at 0.10% of ANE (Figure 7).

The chlorophyll a was 0.10 ± 0.001 mg in PFA (ANOVA; F = 21560.3; *p* < 0.01) and 0.19 ± 0.001 mg (ANOVA; F = 688,291; *p* < 0.01) in PRA; the chlorophyll b was 0.11 ± 0.002 mg in PFA (ANOVA; F = 16,084.42; *p* < 0.01) and 0.30 ± 0.002 mg in PRA (ANOVA; F = 226,096; *p* < 0.01); total chlorophyll was 0.06 ± 0.002 mg (ANOVA; F = 5602.03; *p* < 0.01) in PFA and 0.19 ± 0.002 mg in PRA (ANOVA; F = 117,221; *p* < 0.01); and the total carotenoid content was 0.7 ± 0.001 mg (ANOVA; F = 14,375.03; *p* < 0.01) in PFA and 1.39 ± 0.01 mg (ANOVA; F = 484,563; *p* < 0.01) and PRA experiments (Table 9). The photosynthetic pigment was the lowest in the control in both PFA and PRA of ANE. Pot root application accumulated more of the photosynthetic pigments compared to pot foliar application of ANE (T Test; *p* < 0.000, (Table 10)).

## 3. Discussion

The application of natural organic fertilizer to crop plants is a suitable approach for increasing the yield without any negative impact on the environment. The different concentrations of *Ascophyllum nodosum* extract stimulated the positive growth of *V. aconitifolia* when either applied on the leaves or the roots. The foliar spray of *Ulva reticulate* on *V. radiata* increased the leaf area, shoot length, and root length up to 2%, which thereafter decreased up to 8% [23]. The results of Rayorath et al. [24] also observed that the treatment of *A. nodosum* at low concentration increased the number of leaves in *Arabidopsis*. The extracts of *S. wightii* and *U. lactuca* increased the number of leaves in *Glycine max* [25]. The seaweed treatment increased in the number of leaves in *Abelmoschus esculentus* [26]. The area of the leaves in *Solanum melongena* [27] and in *G. max* [28] was increased with the application of seaweed.

ANE is a good source of nutrients and growth regulators which perform crucial roles in delaying senescence. The cytokinin suppresses the action of ethylene and abscisic acid. The cytokinins retain membrane integrity by inhibiting the activity of the enzymes lipase and lipoxygenase, which lead to a breakdown of the membrane [29]. The use of commercial extract of *A. nodosum* is more beneficial to plants than dried seaweed [30]. The present study concurs with the findings of Tandon et al. [28] who reported an increase in the number of nodules upon treatment with Biozyme on *Glycine max*. Khan et al. [14] also reported an increase in the number of nodules as well as growth of alfalfa. Seaweed based panchagavya treatment increased the number of nodules in *Arachis hypogea* [31]. The increase in the number of nodules in *A. nodosum*-treated plants may be explained on the basis of its composition, especially the presence of the nod-genes promoting factors [14]. The whole seaweed along with highly degraded fucoidan and alginic acid increases the activity of soil micro flora as well as the root system. The oligomers and polysaccharides in *A. nodosum* may play an important role as attractive active elicitors. Badri et al. [32] reported that the plants secrete a variety of chemicals from their roots to attract microorganisms to form a symbiotic relationship. The *A. nodosum* is a potential source of alginic acid and fucoidans. The alginic acid of *A. nodosum* forms high-molecular-weight complexes with the ions in the soil that can retain and absorb moisture, which in turn improve soil aeration. These improve microbial growth and encourage the root systems of plants for better growth [33,34]. Sehrawat et al. [35] performed Amplified Rhizobial DNA Restriction Analysis (ARDRA) of 16S rDNA gene sequences of rhizobial DNA isolated from the control and treated nodules using the forward primer BAC27F (5′-AGA GTT TGA TCC TGG CTC AGG-3′) and reverse primer 1378R (5′ CGG TGT GTA CAA GGC CCG GGA ACG-3′). Two restriction enzymes (HaeIII and MspI) were used to get restriction patterns from the amplified products. Maximum nodulation and enormous rhizobial diversity were recorded in the 0.05% ANE-treated *V. aconitifolia.*

The flavonoids in *A. nodosum* helps in the signaling of legumes–rhizobial interactions. The increase in nodule number may be due to the elicitation by flavonoids, which in turn promote the nodulation process. It was observed that alginate oligosaccharides, which are produced by enzymatic breakdown of alginic acid, mainly extracted from brown algae, improve the soil’s microbial community significantly [36]. Compared to control, the ANE-treated plants had a higher biomass, which may be correlated with a greater number of nodules in the presence of ANE. Moreover, ANE increases nutrient absorption by the roots in terms of increased water and nutrient efficiency that results in the increased plant growth. Moreover, betaines in the ANE may acts as a nitrogen supplier at lower concentrations, whereas at higher concentrations it works as an osmolyte. Khan et al. [14] observed increased shoot and root dry weight in ANE-treated *M. sativa* compared to the control. The increased shoot and root fresh weight as well as dry weight in *Vicia faba* was observed by the application of seaweed extracts [37] Hernández-Herrera et al. [38] observed that the application of extracts of seaweed (*U. lactuca* and *p. gymnospora*) on *Solanum lycopersicum* showed positive effects on the fresh weight of plants compared to the control. Lorenc et al. [39] found the greatest shoot fresh and dry weight following the application of commercial Bio-Algeen in spruce seedlings. The increase in fresh weight, dry weight, shoot length, and root length in *Solanum melongena* was also observed in the presence of brown seaweed (*Stoechospermumm arginatum)*. Mahmoud El-Sharkawy et al. [40] found a higher water content in Alfalfa following the application of seaweed extracts. The low concentration of seaweed liquid fertilizer increased moisture content, leaf area, biomass, and many biochemical parameters in crop plants [8]. Pise et al. [41] studied the effect of three seaweed extracts on *Trigonella foenum-graecum* and found greater dry mass accumulation and moisture content compared to the control. The application of ANE in concentrations higher than 0.01% resulted in a significant change in the number of pods compared to the control. Pre-harvesting spray of *A. nodosum* is known to increase the quantity as well as the quality of crop plants [42,43]. The extract of *Halimeda opuntia* increased *Vicia faba* [37], *S. latifolium* and *Caulerpa racemosa* increased *V. mungo* [44], and *A. nodosum* (Biozyme) increased *Glycine max* [28] yield due to the increase in the number of pods in treated plants. Studies have reported increased photosynthetic pigments following the application of SLF [28,45]. Betaines are important constituents of ANE. The betaine is also known to prevent the degradation of chlorophyll. The three seaweed extracts (*U. fasciata*, *S. ilicifolium*, and *Gracilaria corticata*) significantly increased the chlorophyll and carotenoid content in *T. foenum-graecum* [41]. The results of our work concur with earlier studies that reported an increase in chlorophyll content in crop plants following the treatment of ANE. An increase in chlorophyll ‘a’, chlorophyll ‘b’, total chlorophyll, and carotenoids in *A. esculentus* by *Rosenvigea intricate* [46] and in *V. mungo* by *U. reticulate* [8,23] was also observed. The application of *A. nodosum* was also previously reported to enhance the chlorophyll content in the leaves of *Solanum lycopersicum*, *Hordeum vulgare*, *Phaseolus vulgaris*, *Zea mays*, *V. radiata*, and *Triticum aestivum* [3,47]. Overall, ANE supplied through roots showed better results compared to foliar application of ANE. It is well documented that the leaves, stems, and flowering tissues of plants absorb limited amount of nutrients. This may be due to the site of the application. By foliar application, many essential nutrients of ANE cannot directly enter through the leaves due to the presence of the cuticle. However, in the case of root application, the nutrients are absorbed in to plants via the roots. In addition, the nutrients percolate through the soil and can be easily taken in by root hair and channeled to the other parts of the plant (sink) for metabolic processes.

## 4. Materials and Methods

### 4.1. Preparation of Ascophyllum Nodosum Extract (ANE) and Seed Germination

The seaweed, *A. nodosum* (Trade name: Biovita) was purchased from PI Industries, Udaipur, Rajasthan. This Biovita is 20% natural seaweed extract and 0.25% preservatives. Different concentrations (0.00% (control), 0.01%, 0.02%, 0.05%, 0.10% 0.50%, and 1.00%) of the *A. nodosum* Extract (ANE) were prepared in distilled water. Pure culture of a *Bradyrhizobium* strain, *MB 703* was procured from the department of Microbiology CCS Haryana Agricultural University, Hisar.

The *V. aconitifolia* (Var. RMO 225) was procured from the Department of Genetic and Plant Breeding (Pulses Section) CCSHAU, Hisar, Haryana. The healthy, uniform seeds were surface-sterilized with mercuric chloride (0.1%) and then washed with sterilized distilled water. The seed germination was conducted in plastic pots (30 × 30 cm) filled with 4.0 kg of sterilized river sand. Seven seeds were sown at a depth of 1.0 cm in each pot. Germination was recorded daily for 7 days. After the 5th day of germination, three seedlings/pots were kept for further experiment. Moisture content and temperature of sand were also measured.

### 4.2. Experimental Design and A. nodosum Extract (ANE) Treatment

Two experiments, i.e., pot root application (PRA) and pot foliar application (PFA) were conducted for the present study. The different concentrations (0.00%, 0.01%, 0.02%, 0.05%, 0.10%, 0.50%, and 1.00%) of *A. nodosum* extract (ANE) were applied to the *V. aconitifolia* plants through roots in PRA and on the leaves in PFA experiments. For both PRA and PFA experiments, the surface-sterilized seeds were soaked in different concentrations (0.00%, 0.01%, 0.02%, 0.05%, 0.10%, and 0.50%) of ANE for 12 h. Then the seeds were soaked in a pure culture of Bradyrhizobium species strain MD 703 (10^9^ cells) for 1 h. Then the soaked seeds were sown in pots at a depth of 1 cm, at spacing of 15-20 cm. Three pots from each treatment and three plants/pots for were kept for further study. In the PFA experiment, the ANE was applied after the 3rd day via foliar spray iatn concentrations of 0.00%, 0.01%, 0.02%, 0.05%, 0.10%, and 0.50% of ANE. In total, 5.0 mL ANE/plant was sprayed at a regular interval of 15 days. In the PRA experiment, 5.0 mL of each concentration of ANE was applied to the roots/plant at a regular interval of 15 days. A total of 100 mL of the nitrogen-free Slogar’s solution was given every second evening, alternating with distilled water [48]. The crop was harvested on 80th day.

### 4.3. Growth Parameters and Yield Attributes

The plant growth in terms of shoot length, root length, leaf number and size, nodule number, pod number, and yield were recorded. Fresh and dry weight, and photosynthetic pigments were recorded on the day of uprooting of the plants (80th day). The shoot length (cm) from the region of the collar to the tip was measured on the 15th, 30th, 45th, 60th, and 75th day. The root length (cm) was measured on the 80th day of crop. The leaf number and leaf area were observed on the 30th day of sowing. The numbers of pods and seed yield were recorded at maturity. The nodule numbers were counted upon uprooting on the 80th day. Organic content, water content, and percentage moisture were also observed and calculated by using the following formula:Organic content = Dry weight(1)
Water content = Fresh weight − Dry weight(2)
Moisture % = {(Fresh weight − Dry weight) / Fresh weight} × 100(3)

### 4.4. Photosynthetic Pigments

#### 4.4.1. Chlorophyll Content Estimation

The fresh leaves (1.0 gm) were harvested from treated and control plants. A total of 5.0 mL of 80% acetone was used to grind the leaves. The mixtures were centrifuged for 15 min at 5000 rpm. The supernatant was collected in a storage vial. The pellets were re-extracted with 5.0 mL of 80% acetone. Both the extracts (supernatant) were pooled and used for photosynthetic pigment determination by UV-Visible spectrophotometer at a wavelength of 645 nm and 663 nm. The chlorophyll a, chlorophyll b, and total chlorophyll were estimated using the formulae given below:Chlorophyll a (mg/g.fr. wt.) = {(12.7 × ΔA663 − 2.69 × ΔA645)/(a × 1000 × W)} × V(4)
Chlorophyll b (mg/g.fr. wt.) = {(22.9 × ΔA645 − 4.68 × ΔA663)/(a × 1000 × W)} × V(5)
Total Chlorophyll (mg/g.fr. wt.) = {(20.2 × ΔA645 − 8.02 × ΔA663)/(a × 1000 × W)} × V(6)
where, ΔA = Absorbance at different wavelength

V = Volume of extract (mL)

W = Fresh weight of the sample (g)

#### 4.4.2. Estimation of Carotenoid Content

The extracts obtained above were also used to study the carotenoid content at an absorbance of 480 nm by using the following formula:Carotenoid (µg/g.fr. wt) = ΔA480 + (0.114 × ΔA663) − (0.638 × ΔA645)(7)
where, ΔA = Absorbance at respective wavelength

### 4.5. Statistical Analysis

The experiments were carried out in triplicates. The values were expressed as means ± standard deviation. The results obtained were tested for significance using one way-analysis of variance (One Way-ANOVA). Furthermore, the differences between the treatments were tested by employing the HSD Tukey Test using SPSS software. A *p*-value less than 0.01 was considered as significant.

## 5. Conclusions

Bioactive phytoconstituents in ANE not only promote agricultural productivity, nutrient uptake, and soil properties, but also the enhance the rhizobia community. The present study showed that lower concentrations (0.05% and 0.10%) of ANE were the most effective for capitulation of growth and yield attributes of *V. acontifolia.* It is also suggested that root application of a low concentration of ANE is superior to minimize biofertilizer wastage.

## Figures and Tables

**Figure 1 plants-10-02361-f001:**
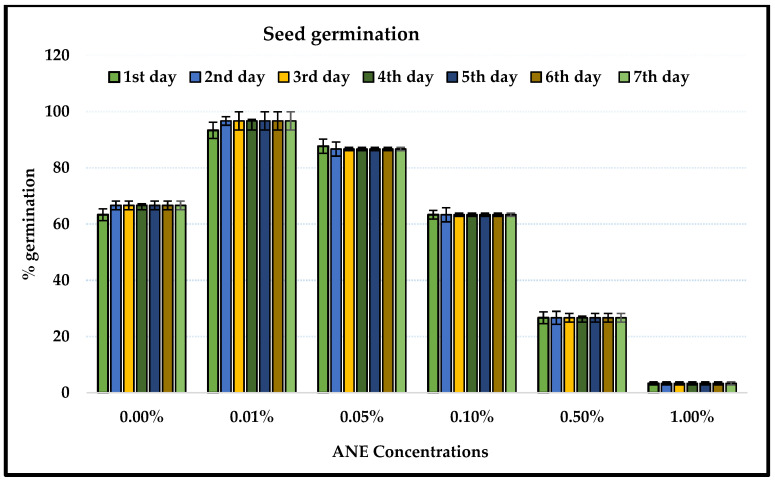
Seed germination of *V. aconitifolia* under different concentrations of *A. nodosum*. Bar indicates mean ± SD (n = 3).

**Figure 2 plants-10-02361-f002:**
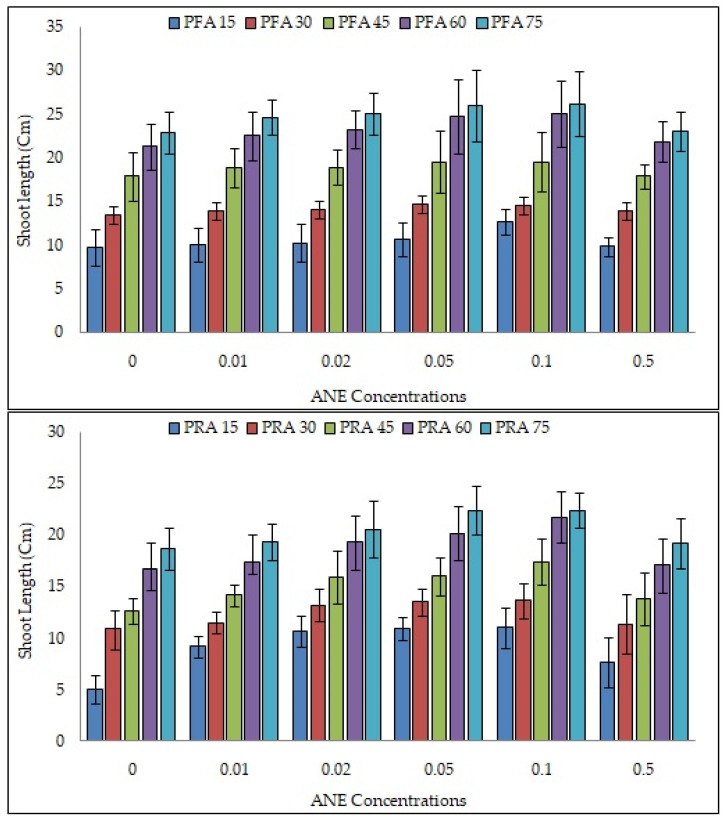
Efficacy of ANE on shoot length after foliar application (PFA) and root application. (PRA) in *V. aconitifolia.* Bar indicates mean ± SD (n = 3).

**Figure 3 plants-10-02361-f003:**
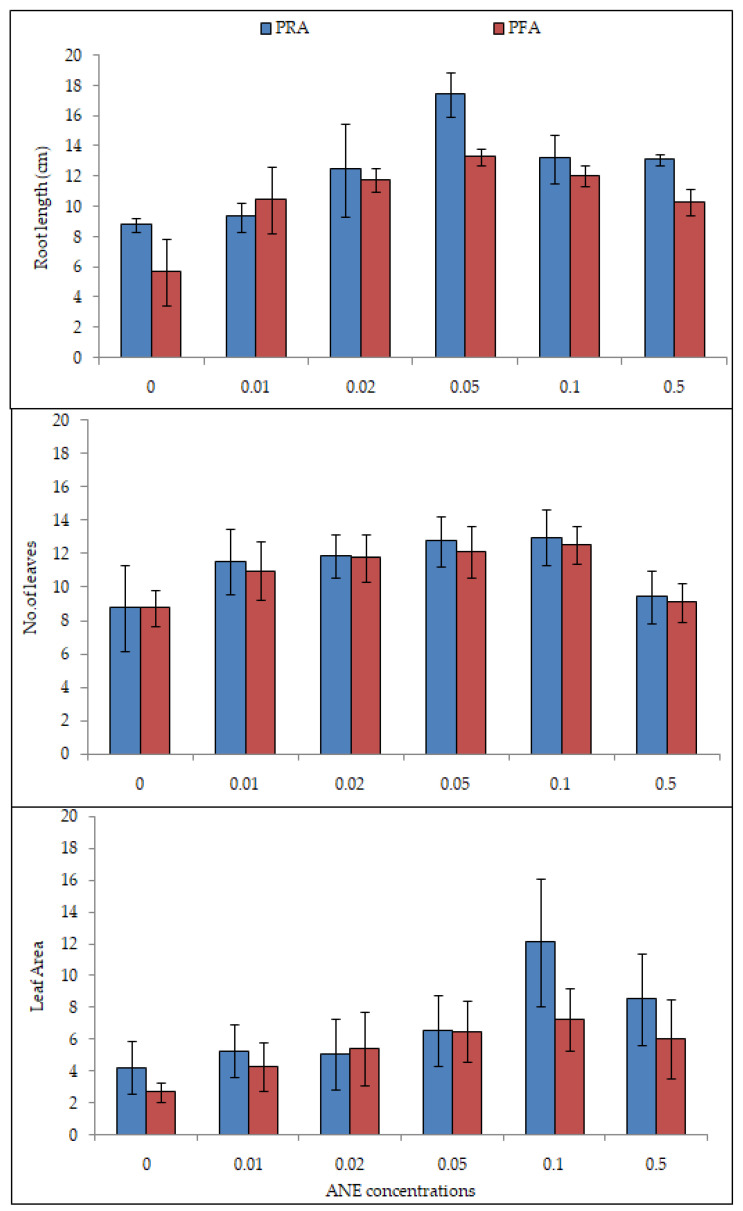
Efficacy of ANE treatment after pot foliar (PFA) and root application (PRA) on root growth, number of leaves, and leaf area on 30th day of growth of *V. aconitifolia.* Bar indicates mean ± SD (n = 3).

**Figure 4 plants-10-02361-f004:**
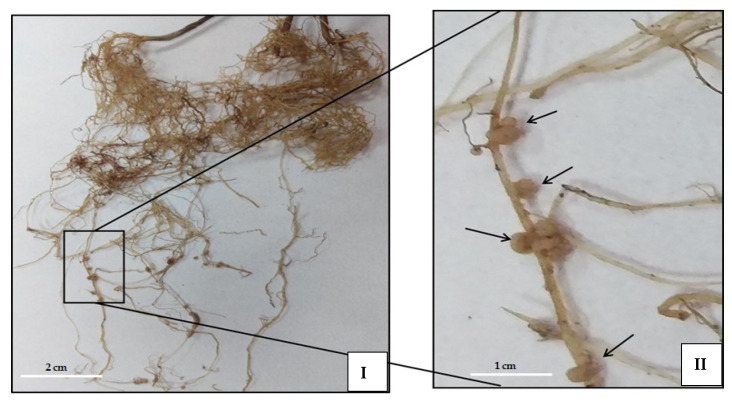
Root nodules at 0.05% ANE treatment in pot root application (PRA) experiment of *V. aconitifolia.* Scale bar is showing 2cm for (**I**), and 1.0 cm (**II**).

**Figure 5 plants-10-02361-f005:**
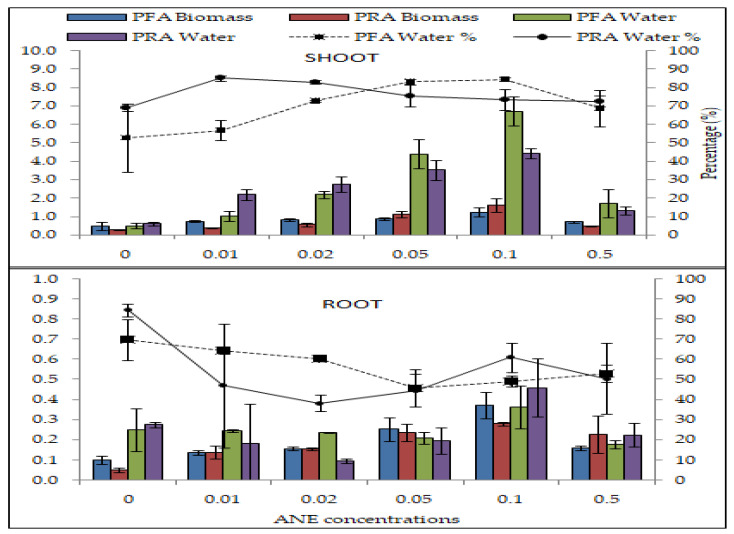
Efficacy of ANE on biomass accumulation in shoots and roots of *V. aconitifolia* in pot foliar (PFA) and pot root application (PRA). Bar indicates mean ± SD (n = 3).

**Figure 6 plants-10-02361-f006:**
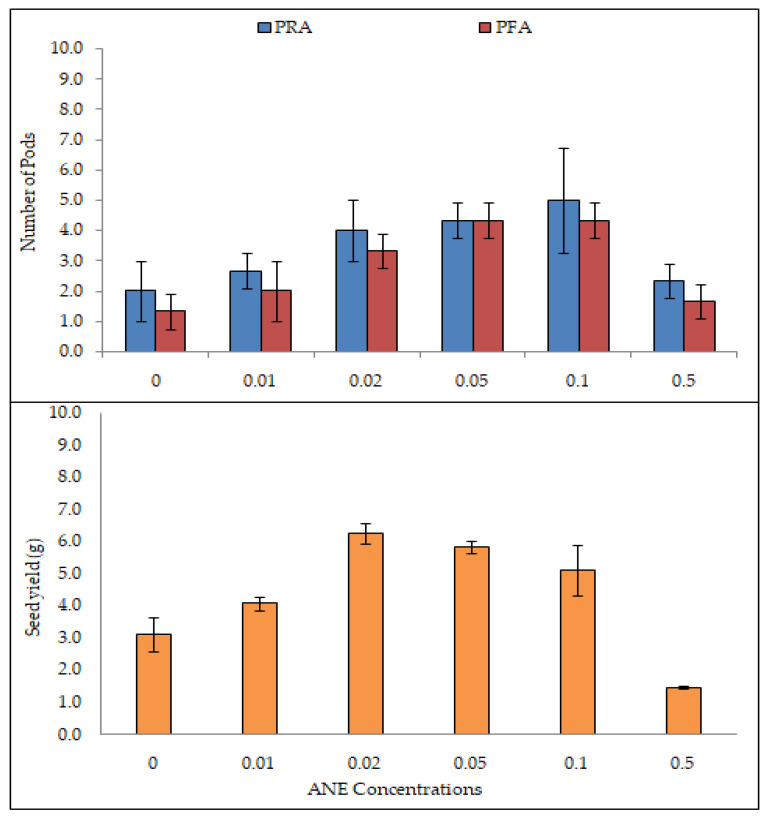
Efficacy of ANE treatments in root application on yield of seeds of *V. aconitifolia.* Bar indicates mean ± SD (n = 3).

**Figure 7 plants-10-02361-f007:**
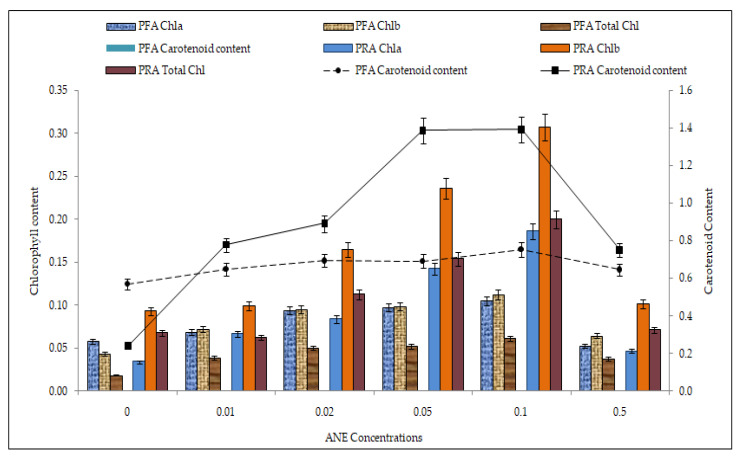
Efficacy of ANE on photosynthetic pigment accumulation. Bar indicates mean ± SD (n = 3).

**Table 1 plants-10-02361-t001:** ANOVA for seed germination with ANE treatments on the 3rd day of germination.

Percent Germination on 3rd Day					
Source	SS	df	MS	F	*p*
Treatment	8706.67	4.00	2176.67	23.32	<0.0001
Error	933.33	10.00	93.33		
Total	9640.00	14.00			

**Table 2 plants-10-02361-t002:** ANOVA for shoot growth at various concentrations of ANE in *Vigna aconitifolia*.

15th Day PRA						15th Day PFA				
Source	SS	df	MS	F	*p*	SS	df	MS	F	*p*
Between Groups	253.88	5	50.78	18.80	2.6 × 10^−10^	53.92	5	10.78	3.24	NS
Within Groups	129.63	48	2.70			159.76	48	3.33		
Total	383.51	53				213.69	53			
**60th Day PRA**						**60th Day PFA**				
Between Groups	176.21	5	35.24	6.64	9 × 10^−5^	103.21	5	20.64	2.19	NS
Within Groups	254.90	48	5.31			453.16	48	9.44		
Total	431.10	53				556.37	53			

**Table 3 plants-10-02361-t003:** ANOVA for root growth upon uprooting after treatment of ANE in PFA and PRA experiments.

PRA						PFA				
Source	SS	df	MS	F	*p*	SS	df	MS	F	*p*
Between Groups	146.32	5	29.26	11.33	0.0003	104.47	5	20.89	10.67	0.0004
Within Groups	30.99	12	2.58			23.50	12	1.96		
Total	177.31	17				127.97	17			

**Table 4 plants-10-02361-t004:** ANOVA for number of leaves of *V. aconitifolia* on 30th day.

PRA						PFA				
Source	SS	df	MS	F	*p*	SS	df	MS	F	*p*
Between Groups	114.22	5	22.84	12.31	<0.0001	137.43	5	27.49	8.43	<0.0001
Within Groups	89.11	48	1.86			156.44	48	3.26		
Total	203.33	53				293.87	53			

**Table 5 plants-10-02361-t005:** ANOVA for leaf area of *Vigna aconitifolia* under different ANE treatments.

PRA						PFA				
Source	SS	Df	MS	F	*p*	SS	df	MS	F	*p*
Between Groups	278.50	5	55.70	1.68	0.21	69.33	5	13.87	1.92	0.16
Within Groups	398.00	12	33.17			86.67	12	7.22		
Total	676.50	17				156.00	17			

**Table 6 plants-10-02361-t006:** ANOVA for the efficacy of ANE via PFA and PRA experiments on biomass accumulation in shoots.

**ANOVA: PRA**															
**Shoot Organic Content**						**Shoot Moisture Content**					**Shoot Moisture Content**				
**Source**	**SS**	**Df**	**MS**	**F**	** *p* **	**SS**	**df**	**MS**	**F**	** *p* **	**SS**	**df**	**MS**	**F**	** *p* **
B/w Grps	0.96	5	0.19	9.94	0.0006	83.7	5	16.74	51.7	<0.0001	2582.74	5	516.55	6.64	0.003
Within Grps	0.23	12	0.02			3.89	12	0.32			933.17	12	77.76		
Total	1.20	17				87.6	17				3515.91	17			
**ANOVA: PFA**															
**Shoot Organic Content**						**Shoot Moisture Content**					**Shoot Moisture Content**				
**Source**	**SS**	**df**	**MS**	**F**	** *p* **	**SS**	**df**	**MS**	**F**	** *p* **	**SS**	**df**	**MS**	**F**	** *p* **
B/w Grps	3.96	5	0.79	26.7	<0.001	29.55	5	5.91	51.36	<0.0001	586.48	5	117.3	8.66	0.001
Within Grps	0.36	12	0.03			1.38	12	0.12			162.48	12	13.54		
Total	4.32	17				30.93	17				748.96	17			

**Table 7 plants-10-02361-t007:** ANOVA showing the efficacy of ANE via PFA and PRA experiments on biomass accumulation in roots.

**ANOVA: PRA**															
**Root Organic Content**						**Root Moisture Content**					**Root Moisture Content**				
**Source**	**SS**	**Df**	**MS**	**F**	** *p* **	**SS**	**df**	**MS**	**F**	** *p* **	**SS**	**df**	**MS**	**F**	** *p* **
B/w Grps	0.15	5	0.03	21.8	<0.003	0.06	5	0.01	3.01	0.054	1311.44	5	262.29	7.12	0.003
Within Grps	0.02	12	0.00			0.05	12	0.0			442.20	12	36.85		
Total	0.17	17				0.11	17				1753.64	17			
**ANOVA: PFA**															
**Root Organic Content**						**Root Moisture Content**					**Root Moisture Content**				
**Source**	**SS**	**df**	**MS**	**F**	** *p* **	**SS**	**df**	**MS**	**F**	** *p* **	**SS**	**df**	**MS**	**F**	** *p* **
B/w Grps	0.10	5	0.02	10.7	0.0004	0.23	5	0.05	4.09	0.02	4132.56	5	826.5	3.54	0.03
Within Grps	0.02	12	0.002			0.13	12	0.01			2802.22	12	233.5		
Total	0.13	17				0.36	17				6934.78	17			

**Table 8 plants-10-02361-t008:** ANOVA for efficacy of ANE on pod number when applied via pot foliar application (PFA) and pot root application (PRA).

PRA						PFA				
Source	SS	Df	MS	F	*p*	SS	df	MS	F	*p*
Between Groups	22.28	5	4.46	4.46	0.02	27.17	5	5.43	12.23	0.0002
Within Groups	12.00	12	1			5.33	12	0.44		
Total	34.28	17				32.5	17			

**Table 9 plants-10-02361-t009:** ANOVA for pot foliar application (PFA) and pot root application (PRA) of ANE on the photosynthetic pigments.

**Total Chl: PRA**						**Total Chl: PFA**				
**Source**	**SS**	**Df**	**MS**	**F**	** *p* **	**SS**	**df**	**MS**	**F**	** *p* **
Between Groups	0.05	5	0.01	117,220.8	<0.0001	0.003	5	0.001	5602.03	<0.0001
Within Groups	0.00	12	0.00			0.000	12	0.000		
Total	0.05	17				0.003	17			
**Total Carotenoid: PRA**						**Total Carotenoid: PFA**				
**Source**	**SS**	**Df**	**MS**	**F**	** *p* **	**SS**	**df**	**MS**	**F**	** *p* **
Between Groups	2.87	5	0.57	484,561.80	<0.0001	0.06	5	0.01	14,375.03	<0.0001
Within Groups	0.00	12	0.00			0.00	12	0.00		
Total	2.87	17				0.06	17			

**Table 10 plants-10-02361-t010:** T test depicting the efficacy of ANE on pigment accumulation in pot foliar application (PFA) and pot root application (PRA) experiments.

T Test	PFAVs PRA		
	T	df	*p*
Chla	−720.465	2	<0.0001
Chlb	−667.701	2	<0.0001
Total Chl	−493.448	2	<0.0001
Carotenoid	−627.176	2	<0.0001

## Data Availability

All data, tables, figures and results in paper are our own and original.

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
