# Peer review of "Potential Use of *Ascophyllum nodosum* as a Biostimulant for Improving the Growth Performance of *Vigna aconitifolia* (Jacq.) Marechal"

_plants, 2021, doi:10.3390/plants10112361_

Round 1

Reviewer 1 Report

Authors have studied the effect of A. nodosum in growth improvement of vigna. following points need due attention:

Author conclude that 0.01% was affective to increase growth but 0.05% improved yield attributes, this is contradictory conclusion. growth promotion and yield improvement should correlate with each other.

English language is so weak and needs a thorough revision. 

experimental design is very much confusing. Soaking can never be considered as foliar or root application, they have used soaking method (priming), and had continued the foliar as well root application method. this clearly contradicts the authors claim of PFA and PRA.

Results are not enough to confirm the beneficial effects of A nodosum, what about the biochemical and molecular parameters.

Was the extract analyzed for the compositions and what components contributed to its function.

Author Response

Answers to Comments-Reviewer 1

No.

Comment

Answer

1

Are the methods adequately described?

Revised

2

Are the results clearly presented?

Improved

3

Are the conclusions supported by the results?

Improved

4

Author conclude that 0.01% was affective to increase growth but 0.05% improved yield attributes, this is contradictory conclusion. growth promotion and yield improvement should correlate with each other.

No, 0.01% was not concluded as the best concentration.  Most of the parameters including yield attributes under study showed that either 0.05% or 0.10% of ANE were better.

5

English language is so weak and needs a thorough revision

Language revised

6

Experimental design is very much confusing. Soaking can never be considered as foliar or root application, they have used soaking method (priming), and had continued the foliar as well root application method. This clearly contradicts the authors claim of PFA and PRA.

Twelve hours of soaking was done to imbibe the seed for better penetration of Rhizobial inoculum for establishing a good symbiotic relationship.

 Ascophyllum nodosum was applied via root and foliar application (PRA and PFA)  to know which is the better way for applying the biofertilizer to get maximum benefit   to the crop without  wastage of the biofertilizer

7

Results are not enough to confirm the beneficial effects of A nodosum, what about the biochemical and molecular parameters.

Research in this paper only focused on the growth performance and yield attributes under the influence of ANE.

Papers for biochemical studies (reference no 15 &16) and Molecular characterization (reference no 48) are already published and cited in this manuscript.

8

Was the extract analyzed for the compositions and what components contributed to its function?

No, the extract was not analyzed for its components.

Research work was planned as per available literature, that growth regulators and other nutrients in ANE play an important role for the plant growth. i.e Components like

Alginic acids, fucoidan,  betain etc. increase the activity of soil microflora and root system thus increase the productivity, though high conc of betain act as osmolytes.

Reviewer 2 Report

The manuscript deals with the Ascophyllum nodosum extract and its effect on different growth parameters of Vigna aconitifolia.

The manuscript meets expectations of Plants.

Nevertheless this manuscript lack of originality and novelty.

Moreover , the manuscript is presented as a listing and do not bring a mechanistic approach.

Author Response

Answers to Comments-Reviewer 2

No.

Comment

Answer

1

Does the introduction provide sufficient background and include all relevant references

Improved

2

Is the research design appropriate?

Improved

3

Are the methods adequately described?

Revised

4

Are the results clearly presented?

Improved

5

Are the conclusions supported by the results?

Improved

The whole MS is critically checked and thoroughly revised

Reviewer 3 Report

Nidhi et al., submitted a paper to Plants entitle ‘Potential use of Ascophyllum nodosum as a Biostimulant for Improving the Growth Performance of Vigna aconitifolia (Jacq.) Marechal’. Vigna aconitifolia is a drought-resistant legume, commonly grown in arid and semi-arid regions of India. In this study, the authors treated seaweed extract as natural bioactive fertilizer. They treated V. aconitifolia with two application methods (Root Application; PRA and Foliar Application; PFA), and 7 concentrations (0.00%, 0.01%, 0.02%, 0.05%, 0.10%, 0.50 % and 1.00%) to record the plant responses. The authors provide an interesting study here.

I have some major questions for the authors:

  1. When authors added 0.5% and 1% of seaweed extract to the seeds, the germination rate was reduced. Authors should provide the ion content and concentration in the seaweed extract?
  2. As my understanding, the responses of foliar application faster than root application, in figure 2 the root application faster than foliar application. Can authors explain the possible reasons?
  3. The quality of figures are worse, the quality and style of Figs. should be largely improved.

I have some minor suggestions for the authors:

  1. Figure 2 .legend typo: pplication should be application.
  2. The resolution of figures is not good enough. Some Figs. are out of shape.
  3. The color of bars should be reconsidered. For example, the gold color bar in Fig.10 is dazzling. Green bars in Fig. 9 also inappropriate (and the gray background).
  4. Scale bars should be included in Figs. 3, 4 and 6.
  5. Please check the reference style.

Author Response

Answers to Comments-Reviewer 1

No.

Comment

Answer

1

Does the introduction provide sufficient background and include all relevant references

Examine critically and improved

1

Are the methods adequately described?

Checked and revised thoroughly

2

Are the results clearly presented?

Examine again cortically and edited

3

Are the conclusions supported by the results?

Improved

4

When authors added 0.5% and 1% of seaweed extract to the seeds, the germination rate was reduced. Authors should provide the ion content and concentration in the seaweed extract?

Though the ion content didn’t studied in this research but as per available literature that Fucoidan and alginic acid Betain at higher concertation may be responsible for reducing the seed germination although low concentration increased the germination percent of V. acontifolia

5

As my understanding, the responses of foliar application faster than root application, in figure 2 the root application faster than foliar application. Can authors explain the possible reasons?

It may be explained on the basis of composition of ANE especially:

1. Fucoidan and alginic acid increases the activity of microflora

2. nod genes promoting factors in ANE

3. oligomers in ANE attracts the elicitors

4.Nutients in ANE are absorb by roots which increases the growth of plants

5.  Betain in ANE at low concertation supply nitrogen

6

The quality of figures are worse, the quality and style of Figs. should be largely improved.

Improved

7

Figure 2 .legend typo: pplication should be application

corrected

8

The resolution of figures is not good enough. Some Figs. are out of shape.

Modified

9

The color of bars should be reconsidered. For example, the gold color bar in Fig.10 is dazzling. Green bars in Fig. 9 also inappropriate (and the gray background).

Changed

10

Scale bars should be included in Figs. 3, 4 and 6

Scale bars have been included in the mentioned figures

11

Please check the reference style.

Checked and edited

Answers to Comments-Reviewer 1

No.

Comment

Answer

1

Does the introduction provide sufficient background and include all relevant references

Examine critically and improved

1

Are the methods adequately described?

Checked and revised thoroughly

2

Are the results clearly presented?

Examine again cortically and edited

3

Are the conclusions supported by the results?

Improved

4

When authors added 0.5% and 1% of seaweed extract to the seeds, the germination rate was reduced. Authors should provide the ion content and concentration in the seaweed extract?

Though the ion content didn’t studied in this research but as per available literature that Fucoidan and alginic acid Betain at higher concertation may be responsible for reducing the seed germination although low concentration increased the germination percent of V. acontifolia

5

As my understanding, the responses of foliar application faster than root application, in figure 2 the root application faster than foliar application. Can authors explain the possible reasons?

It may be explained on the basis of composition of ANE especially:

1. Fucoidan and alginic acid increases the activity of microflora

2. nod genes promoting factors in ANE

3. oligomers in ANE attracts the elicitors

4.Nutients in ANE are absorb by roots which increases the growth of plants

5.  Betain in ANE at low concertation supply nitrogen

6

The quality of figures are worse, the quality and style of Figs. should be largely improved.

Improved

7

Figure 2 .legend typo: pplication should be application

corrected

8

The resolution of figures is not good enough. Some Figs. are out of shape.

Modified

9

The color of bars should be reconsidered. For example, the gold color bar in Fig.10 is dazzling. Green bars in Fig. 9 also inappropriate (and the gray background).

Changed

10

Scale bars should be included in Figs. 3, 4 and 6

Scale bars have been included in the mentioned figures

11

Please check the reference style.

Checked and edited

Round 2

Reviewer 1 Report

Authors should provide more data to support the findings. 

only 2 to 3 parameters are not enough.

Author Response

Author responses to editor/reviewer comments

#Reviewer 1

 (Plants-1361937)

No.

Comment

Reply

1.

Introduction ------must be improved

Introduction part has been completely revised.

2.

Research design -----must be improved

Research design has been improved accordingly.

4

Are the methods adequately described?----improved

Revised thoroughly and described adequately.

5

Are the results clearly presented? ------Must be improved

Results portion has been rechecked and improved.

6.

Are the conclusions supported by the results? ------------------Must be improved

Conclusion part   completely modified according to the supportive results.

The whole MS is critically checked and thoroughly revised.

The reviewer and editors comments are reasonable, and we have corrected the MS in accordance with the comments and suggestions. A thorough internal reviews was performed in the whole MS, changes highlighted in Track Change Format supplied MS. We are thankful to learned reviewer for giving critical insights, leading to substantial improvement in the manuscript, we hope the response meets the reviewer and editor approval.

Reviewer 2 Report

Dear Authors,

Thank you for your good work ad for considering all remarks.

Author Response

Author responses to editor/reviewer comments

#Reviewer 2

 (Plants-1361937)

No.

Comment

Answer

1

Introduction ---------can be improved

Introduction part has been completely revised.

2.

The methods--------- can be improved

Revised thoroughly and described adequately.

3.

Are the conclusions supported by the results?

Conclusion part   completely modified according to the supportive results.

 The whole MS is critically checked and thoroughly revised

The reviewer and editors comments are reasonable, and we have corrected the MS in accordance with the comments and suggestions. A thorough internal reviews was performed in the whole MS, changes highlighted in Track Change Format supplied MS. We are thankful to learned reviewer for giving critical insights, leading to substantial improvement in the manuscript, we hope the response meets the reviewer and editor approval.

Reviewer 3 Report

Scale bars should be included in Figs. 3, 4 and 6

Author Response

Author responses to editor/reviewer comments

#Reviewer 3

 (Plants-1361937)

No.

Comment

Reply

1

 Introduction ----can be improved

Introduction part has been completely revised.

2

Research design---can be improved

Research design has been improved accordingly.

3

Methods adequately described-------can be improved

Revised thoroughly and described adequately.

4

Results --- can be improved

Results portion has been rechecked and improved.

5

Are the conclusions supported by the results?

Conclusion part   completely modified according to the supportive results.

6

Scale bars should be included in Figs. 3, 4 and 6

Scale bars added in the mentioned figures.

The whole MS is critically checked and thoroughly revised.

The reviewer and editors comments are reasonable, and we have corrected the MS in accordance with the comments and suggestions. A thorough internal reviews was performed in the whole MS, changes highlighted in Track Change Format supplied MS. We are thankful to learned reviewer for giving critical insights, leading to substantial improvement in the manuscript, we hope the response meets the reviewer and editor approval.

Round 3

Reviewer 1 Report

Data presented in paper is so little (only pigments) and cannot be sufficient to justify the hypothesis, authors must do some more biochemical and molecular parameters.

Author Response

Comment: English language and style are fine/minor spell check required

Reply: English language has been improved and corrected spelling throughout the manuscript.

Comment: Data presented in paper is so little (only pigments) and cannot be sufficient to justify the hypothesis, authors must do some more biochemical and molecular parameters.

Reply: For the objective of this present study, we used Ascophyllum nodosum as a Biostimulant for Improvement of the Growth Performance in Vigna aconitifolia in which growth perameters are positively correlated with the pigmentation as we have done this work in this study. Regarding molecular work, in our earlier published study of Nidhi et al. [48] we have already studied that amplification of 16S rDNA gene sequences in rhizobial DNA was carried out by using the forward primer BAC27F (5’-AGA GTT TGA TCCTGGCTC AGG-3’) and reverse primer 1378R (5’ CGG TGT GTA CAA GGC CCG GGA ACG-3’). The restriction enzymes (HaeIII and MspI) were used to cut the rhizobial ge-
nomic DNA. Different restriction patterns were observed when amplified products cut with HaeIII and MspI.

Round 4

Reviewer 1 Report

No comments now